# Clinical Usefulness of Speckle-Tracking Echocardiography in Patients with Heart Failure with Preserved Ejection Fraction

**DOI:** 10.3390/diagnostics13182923

**Published:** 2023-09-12

**Authors:** Yixia Lin, Li Zhang, Xiaoqing Hu, Lang Gao, Mengmeng Ji, Qing He, Mingxing Xie, Yuman Li

**Affiliations:** 1Department of Ultrasound Medicine, Union Hospital, Tongji Medical College, Huazhong University of Science and Technology, Wuhan 430022, China; linyixia@hust.edu.cn (Y.L.); zli429@hust.edu.cn (L.Z.); huxq@hust.edu.cn (X.H.); m202176058@hust.edu.cn (L.G.); jimengmeng97@163.com (M.J.); hqingedu@163.com (Q.H.); 2Clinical Research Center for Medical Imaging in Hubei Province, Wuhan 430022, China; 3Hubei Province Key Laboratory of Molecular Imaging, Wuhan 430022, China

**Keywords:** speckle-tracking echocardiography, heart failure with preserved ejection fraction

## Abstract

Heart failure with preserved ejection fraction (HFpEF) is defined as HF with left ventricular ejection fraction (LVEF) not less than 50%. HFpEF accounts for more than 50% of all HF patients, and its prevalence is increasing year to year with the aging population, with its prognosis worsening. The clinical assessment of cardiac function and prognosis in patients with HFpEF remains challenging due to the normal range of LVEF and the nonspecific symptoms and signs. In recent years, new echocardiographic techniques have been continuously developed, particularly speckle-tracking echocardiography (STE), which provides a sensitive and accurate method for the comprehensive assessment of cardiac function and prognosis in patients with HFpEF. Therefore, this article reviewed the clinical utility of STE in patients with HFpEF.

## 1. Introduction

Heart failure (HF) is a complex clinical heterogeneous syndrome characterized by the reduced capacity of the heart to pump blood, resulting in insufficient cardiac output to meet the body’s metabolic demands [1,2]. HF is also the terminal stage of various cardiovascular diseases, with symptoms (e.g., breathlessness, orthopnea, and fatigue) and signs (e.g., increased jugular venous pressure, hepatojugular reflux, and peripheral oedema) caused by cardiac abnormalities in the structure and/or function [3,4,5,6]. At present, left ventricular (LV) ejection fraction (LVEF) obtained via echocardiography remains a key indicator for the diagnosis and risk stratification of HF patients. However, LVEF has high variability and low repeatability. In various guidelines, the critical values of LVEF in different HF subtypes are different, and there is still a lack of definition of the normal range [7,8]. According to both European and American guidelines, HF is divided into three subtypes: heart failure with preserved ejection fraction (HFpEF, LVEF  ≥ 50%), heart failure with mid-range ejection fraction (HFmrEF, LVEF  ≥  40 and  < 50%), and heart failure with reduced ejection fraction (HFrEF, LVEF  <  40%) [6,9,10]. It is estimated that around 64.3 million people worldwide suffer from HF, of which, HFpEF is the most common subtype, accounting for more than half of all HF patients. The prevalence and incidence of HFpEF are growing with the aging of the population and the increasing prevalence of metabolic syndromes such as hypertension, obesity, and diabetes mellitus [11,12,13,14]. According to various etiologies, HFpEF can be divided into five subtypes: vascular-related HFpEF (hypertension, coronary artery disease, and coronary-microvascular-dysfunction-related HFpEF), cardiomyopathy-related HFpEF (HFpEF patients with hypertrophic cardiomyopathy or infiltrative cardiomyopathies, etc.), right-heart- and pulmonary-related HFpEF (HFpEF resulting from pulmonary hypertension with or without right ventricular dysfunction), valvular- and rhythm-related HFpEF (HFpEF due to valvular disease and atrial fibrillation), and extracardiac-disease-related HFpEF (HFpEF related to metabolic diseases, diseases that often cause high output states and chronic kidney disease, radiotherapy for cancer, etc.) [15]. Patients with HFpEF have LV diastolic dysfunction, increased filling pressure, and normal or slightly impaired systolic function, but LVEF within the normal range. Compared with HFrEF, the diagnosis of HFpEF is more challenging because of the normal LVEF [6]. The recent European Society of Cardiology (ESC) guidelines defined HFpEF as the presence of symptoms and/or signs of HF; preserved LVEF (LVEF > 50%); elevated levels of natriuretic peptide: brain natriuretic peptide (BNP) > 35 pg/mL and/or N-terminal pro-brain natriuretic peptide (NT-proBNP) > 125 pg/mL; and the evidence of diastolic dysfunction and/or structure heart disease (LV hypertrophy; left atrial (LA) enlargement) [6].

In general, objective evidence of a cardiac structural and functional abnormality is often obtained via imaging, and echocardiography is the preferred method for the clinical evaluation of cardiac structure and function at present [16]. Though LVEF is the most commonly used traditional echocardiographic parameter to assess cardiac performance and is readily available in daily practice, it is not sensitive to slight changes in LV systolic function [17,18,19,20]. The symptoms and signs of HFpEF patients are non-specific, and the LVEF is within normal range, which limits the estimation of cardiac function via conventional echocardiographic parameters. With the development of new techniques such as myocardial deformation imaging, early myocardial dysfunction can be identified before LVEF decreases to guide the diagnosis of HFpEF [19,21,22,23]. The early and accurate identification of cardiac dysfunction is essential for risk stratification, clinical management, and prognostic improvement in HFpEF patients. Myocardial strain obtained via cardiac magnetic resonance (CMR) and speckle-tracking echocardiography (STE) has been proven to be a sensitive parameter for the early detection of subclinical cardiac dysfunction, providing a new index for cardiac function evaluation in HFpEF patients [17,18,24,25]. CMR is the gold-standard imaging modality for quantifying volume and EF [26]. Furthermore, CMR allows for the accurate assessment of the structural changes in HFpEF patients, such as LA enlargement, LV hypertrophy and mitral inflow, and pulmonary venous velocity [26]. More recently, CMR has been used to describe the specific tissue composition of myocardial tissue in HFpEF patients through non-invasive methods, such as edema, fat, iron overload, and focal fibrosis [27]. CMR feature tracking imaging can accurately assess myocardial deformation and help find potential causes of HFpEF [26,27]; however, CMR has a variety of contraindications, such as metal implantation, a long scanning time, and being inappropriate for individuals with claustrophobia, which limits the application of CMR in clinical practice [25]. Currently, echocardiography remains the preferred imaging examination technique for the initial evaluation of HFpEF patients, and STE has emerged as a potential technique for evaluating LV function [28]. Therefore, the purpose of this review was to summarize the clinical usefulness of STE in HFpEF patients.

## 2. Myocardial Deformation Imaging

Myocardial strain is defined as the deformation of the myocardium that occurs during the cardiac cycle, which can be observed via various imaging methods [22,29]. The first non-invasive measurement of myocardial strain in human history was obtained via CMR tracing in the late 1980s [30]. Actually, at the beginning of the 1980s, some scholars proposed the method of applying echocardiography to estimate strain [31,32,33]. It was not until the 1990s that one-dimensional myocardial strain assessment in a specific direction via echocardiography was clinically achieved through the post-processing of tissue Doppler imaging (TDI) data. Although TDI is readily available, its main disadvantage is that it is angle dependent, which only allows for the rapid assessment of myocardial strain when left ventricular walls align with the direction of the scan lines [22,34]. With the development of digitization and the improvement in the computation power of computers, two-dimensional STE (2D-STE) became a reliable tool in clinical routine in the early 2000s [29]. The continuous development of echocardiography promoted the emergence of three-dimensional STE (3D-STE). Three-dimensional STE is a novel imaging technique, which overcomes the inherent limitations of 2D-STE by tracking myocardial motions in 3D space [35,36]. Advances in STE allow myocardial deformation imaging to be used as a powerful and valuable clinical quantitative technique to accurately estimate subclinical myocardial dysfunction.

### 2.1. Two-Dimensional Speckle-Tracking Echocardiography

Speckles are spot echoes generated by the phenomena of reflection, scattering, and interference when the ultrasonic beam passes through myocardial fibers shorter than the incident ultrasonic wavelength. Speckles composed of approximately 20 to 40 pixels do not represent the tissue structure but organizational movement. They can move synchronously with organization within a limited time and distance and can maintain the relative stability of morphology, exhibiting specific features for tracking [34,36]. Consequently, speckles can be tracked consecutively frame to frame by dedicated software during the cardiac cycle, and angle independence is achieved by the sum of the absolute difference specific algorithms [37,38,39,40,41].

On the basis of the traditional 2D echo grey scale, 2D-STE uses automatic algorithms to track position variations of speckles in different directions of the region of interest (ROI) frame by frame to quantitatively calculate the deformational parameters of myocardial fibers in different directions (Figure 1) [23,42,43]. 2D-STE is an angle-independent technique that is unaffected by the movement of the adjacent myocardium segments and the whole heart. It allows for the quantitative analysis of the myocardial deformation in all directions (e.g., longitudinal, radial, and circumferential planes) [42,44,45]. A large number of studies have demonstrated that 2D-STE has been widely used in clinical practice and can detect subclinical myocardial dysfunction sensitively and accurately in the early stages of various cardiac diseases [46,47,48]. Therefore, the ability of strain parameters to diagnose disease and assess prognosis may be superior to conventional echocardiographic parameters [49]. Myocardial deformation resulting from real-time cardiac motion is complex, involving multidirectional deformation, while 2D-STE can only measure strain in two directions at a time [50]. Consequently, 2D-STE has a few inherent limitations: it obtains strain parameters based on the assumption that speckles move in the 2D imaging plane and are adequately tracked during the cardiac cycle, which could not accurately track the deformation in all directions. In addition, 2D strain parameters are obtained from multiple planes of different cardiac cycles, leading to the non-repeatability of measured strain data in the condition of arrhythmia. Moreover, image acquisition is affected by “out-of-plane” motion and foreshortened views, which may make it difficult to assess the true myocardial movement of the heart [51,52,53]. 3D-STE, as a newly developed technology, represents a further advance in myocardial deformation imaging and overcomes the limitations of 2D-STE [34,54].

### 2.2. Three-Dimensional Speckle-Tracking Echocardiography

The development of ultrasound scanner technology with the ability to obtain real-time full volume imaging of the left ventricle has prepared the ground for the emergence and advance of 3D-STE [55]. 3D-STE, based on 2D-STE and real-time full volume imaging, is a novel, non-invasive technology for myocardial deformation assessment, and it has achieved continuous tracking of speckles, frame by frame in 3D space, by the analysis of an acquired full volume dataset, thus reflecting the real movement of the myocardium [55,56]. Speckle tracking is accomplished by block matching, a process described as matching and searching natural acoustic marked cubes with specific 3D patterns within each ROI through a specific algorithm during the cardiac cycle [56,57,58,59]. Because the blocks are followed in a 3D full volume, they can be tracked in any direction, and the multidirectional components of the strain can be simultaneously analyzed by post-processing analysis, providing additional deformation parameters (such as area strain) and the comprehensive evaluation of cardiac function in a three-dimensional acquisition [60,61]. Although 3D-STE is a technique still undergoing technological development, a large number of studies have proven its feasibility, reliability, and potential to overcome the inherent limitations of 2D-STE [36,58,60,62,63]. As is known to all, the spatial orientation of myofibers is complex, and the nature of myocardial mechanics is a 3D phenomenon. 3D-STE allows for the simultaneous measurement of strain in multiple directions in a single acquisition, avoiding the foreshortening of apical views and out-of-plane motion of speckles, having become a more useful tool for analyzing the complexity of cardiac mechanics [34,54,56]. 3D-STE allows all deformation parameters to be acquired at one time (Figure 2), shortening the time of image acquisition and analysis and overcoming errors caused by heart rate variability resulting from multiple acquisitions, such as 2D-STE [36,60,64]. However, 3D datasets for analysis require sufficient temporal and spatial resolution, which is achieved by ECG gating. ECG-gated multi-beat 3D acquisition requires patients to hold their breath for at least four cardiac cycles and a regular heart rhythm, which limits the application of 3D-STE in specific conditions such as arrhythmia. However, technical developments have enabled real-time single-beat full volume acquisition, allowing 3D-STE to analyze patients with arrhythmia and inability to hold their breath, at the cost of reduced spatial resolution [65]. In conclusion, compared with traditional echocardiography and 2D-STE, 3D-STE can rapidly, accurately, and comprehensively evaluate the global and regional cardiac function of HFpEF patients.

## 3. Clinical Application of Speckle-Tracking Echocardiography in Patients with HFpEF

### 3.1. Left Ventricular Function

LV myocardial structure and movement patterns are complex. LV myocardial fibers consist of three layers: subendocardial, middle, and subepicardial. The subepicardial myocardial fibers are longitudinally oriented in the right-handed helix; the mid-myocardial fibers are circumferentially oriented; and the subendocardial fibers are longitudinally oriented in the left-handed helix, which allows the LV movement to be divided into longitudinal shortening, radial thickening, and circumferential movement [17,66]. Therefore, the LV strain is described as myocardial deformation occurring in the longitudinal, radial, and circumferential planes during the cardiac cycle, corresponding to the global longitudinal strain (GLS), global radial strain (GRS), and global circumferential strain (GCS), respectively [17,67,68]. The subendocardial myocardium is most sensitive to ischemia and is the first to undergo dysfunction during myocardial ischemia, namely, the first to change GLS.

In the past, it was considered that patients with HFpEF had LV diastolic dysfunction and normal systolic function. However, recent studies have found that the symptoms of HFpEF patients are not completely caused by diastolic dysfunction, and patients already have systolic dysfunction in the early stage of HFpEF [69,70,71]. The longitudinal myocardium contracts in coordination with the circular myocardium, and ventricular torsion compensation increases, keeping LVEF within the normal range in HFpEF patients [6]. Consequently, the early identification of LV systolic dysfunction provides a basis for risk stratification and clinical management in patients with HFpEF. Hashemi et al., used CMR to evaluate LV function in HFpEF patients, and the results showed that there was no significant difference in LVEF between controls and HFpEF patients, but GLS and GCS were significantly impaired in HFpEF patients. The LV septum was found to be the most affected location by regional strain analysis [25]. However, the high price, long scanning time, and inability to be used in patients with metal implants may limit the wide application of CMR in clinical practice. STE remains a promising technique for the evaluation of LV function. Liu et al., found that LVGLS obtained by 2D-STE was significantly lower in patients with HFpEF compared with healthy controls, while LVEF was not significantly different, suggesting the higher sensitivity of LVGLS over LVEF for the detection of LV dysfunction [72]. The research by Smith et al., suggested that subendocardial and subepicardial LVGLS of the basal, middle, and apical segments is significantly reduced in HFpEF patients, indicating that HFpEF not only affects the subendocardial myocardial fibers but all layers of the myocardium [73]. Kosmala et al., used 2D-STE to investigate the cardiac function in 207 symptomatic HFpEF subjects and 60 HFpEF asymptomatic patients, as well as exploring the predictors of adverse outcomes in patients with HFpEF. Their study showed that LVGLS was significantly decreased in both symptomatic and asymptomatic patients. Receiver operator characteristic (ROC) curves showed that LVGLS during exercise (AUC 0.78) could predict symptomatic HFpEF most accurately, and its predictive value was better than that of E/e’ and LVEF [74]. In a prospective investigation of HFpEF patients followed for three years, Wang et al., proved that impaired LVGLS was associated with adverse events, but only the reduction in LVGLS during exercise was an independent predictor of adverse clinical outcomes, showing that LVGLS is of great value in the prognostic evaluation of HFpEF patients [75]. However, the changes in GCS and GRS in patients with HFpEF and their effects on prognosis are contradictory [66]. Some studies showed that there was no significant difference in GCS in patients with HFpEF compared with healthy controls and that GCS was not associated with the occurrence of adverse outcome events in patients. However, other studies reported a reduction in GCS in patients with HFpEF [73,76,77]. Among them, Smith et al., showed that the GCS of subendocardial and subepicardial LV basal, middle, and apical segments in HFpEF patients was significantly reduced, which again confirmed that the LV full-thickness myocardial wall in HFpEF patients was affected [73]. GRS is rarely investigated in patients with HFpEF. Studies observed the absence of difference in GRS between healthy controls and HFpEF patients, although other research suggested a significantly reduction in GRS in HFpEF patients [75,78,79]. Some studies revealed that GRS was not associated with the occurrence of adverse events in HFpEF patients, but additional studies are needed to explore the effect of GRS on prognosis [75]. In recent years, 3D-STE has been increasingly used in patients with HFpEF and is of great value for the comprehensive evaluation of cardiac function. Fan et al., applied 3D-STE to analyze LV function in patients with and without HFpEF; they reported that the area strain (AS) of the LV was significantly impaired in HFpEF patients and negatively correlated with LVEF, indicating that the change in AS occurred sooner than that in LVEF [80]. Luo et al., used 3D-STE to measure the three-dimensional strain of the left ventricle; their results demonstrated that GLS, GCS, GRS, and AS were progressively impaired in HFpEF patients and that AS combined with GLS and GCS was a predictor of LV systolic dysfunction [81].

### 3.2. Left Atrial Function

LV diastolic dysfunction and increased LV filling pressure in HFpEF patients lead to ineffective LA emptying and increased LA volume and afterload, finally resulting in LA remodeling and function impairment. LA enlargement is a recognized indicator of LV diastolic dysfunction and an independent predictor of poor prognosis in patients with HFpEF [82]. The LA volume index (LAVI) is an essential part of the evaluation of LV diastolic function and is an integral component of the diagnostic criteria of HFpEF [83,84]. LA dysfunction usually precedes the occurrence of LA remodeling and plays an important role in evaluating patients’ prognosis [85,86,87]. According to the different roles of the left atrium in different phases of the cardiac cycle, LA function is divided into reservoir, conduit, and booster functions. LA deformation in different phases was quantified by STE to evaluate LA function. LA peak strain occurs during ventricular contraction, representing reservoir function. LA conduit strain appears during LV early diastole, representing passive LA emptying. LA booster strain appears during LA systole, representing active LA emptying [85,88]. The impaired of LA peak strain is associated with LV systolic and diastolic dysfunction and is highly correlated with LVGLS. Left ventricular systolic longitudinal dysfunction leads to a reduction in displacement from the mitral valve to the apex and thus a reduction in passive LA stretch [89]. Santos et al., used 2D-STE to compare 135 HFpEF patients with sinus rhythm and 40 healthy controls and found the impairment of LA reservoir, conduit, and booster strain in HFpEF patients, as well as the fact that LA reservoir dysfunction was independent of LA size. In addition, they reported that patients with lower LA peak strain had a higher rate of heart failure hospitalization and worse left ventricular systolic function, indicating that LA dysfunction is related to the severity and pathophysiology of HFpEF patients [90]. Using 2D-STE, Morris et al., revealed that the sensitivity of LA strain in the diagnosis of early LA dysfunction in HFpEF patients was higher than that of LAVI, confirming that LA strain is a reliable indicator of LA dysfunction. Moreover, LA strain abnormalities were still significantly associated with worse New York Heart Association (NYHA) class and a higher risk of hospitalization for HF after adjusting for gender, age, and LAVI [91]. Tells et al., simultaneously performed right heart catheterization and echocardiography on the study subjects and found impaired LA peak and booster strain in patients with HFpEF. Further, they investigated the relationship between LA strain and hemodynamics in HFpEF, and the results showed that LA peak strain was negatively correlated with pulmonary capillary wedge pressure (r = −0.64, *p* < 0.001) and that LA booster strain was positively correlated with pulmonary capillary wedge pressure (r = 0.72, *p* < 0.001). The study also found that when LA peak strain was less than 33%, the sensitivity and specificity of non-invasive diagnosis of HFpEF were 87% and 77%, respectively. In summary, LA strain can be used as a non-invasive indicator to evaluate the pressure of the cardiac chamber in HFpEF patients [92]. Freed et al., measured LA strain parameters in HFpEF patients by 2D-STE and followed up these patients. They found that LA function was impaired in HFpEF patients at all phases and LA strain (especially LA reservoir strain) was an independent predictor of adverse outcomes. Moreover, LA reservoir strain was significantly associated with reduced exercise capacity, decreased cardiac output, and the increased risk of poor prognosis. Therefore, therapies that enhance LA function may be beneficial for HFpEF patients [93]. Because of the out-of-plane motion of the speckles and the segmentation of LA following the segmentation of LV, LA structure and function cannot be comprehensively evaluated by 2D-STE. The newly developed 3D-STE more objectively quantifies LA deformation and function and compensates for the shortcomings of 2D-STE. Liu et al., evaluated the LA function in 43 HFpEF patients and 18 healthy subjects. The results of this study showed that LA strain reduction in HFpEF patients with normal LA size predominantly occurred in the middle part of the LA, and strain of basal and roof levels also significantly decreased with the increase in LA size. The reproducibility of strain at the LA middle level was satisfactory (ICC > 0.8), which is an ideal indicator to evaluate LA function. In addition, they also reported that LA reservoir, conduit, and booster function were significantly impaired in HFpEF patients, and these changes were more significant in patients with LA enlargement [94].

### 3.3. Right Ventricular Function

For a long time, the focus of attention in HFpEF patients has been left heart function, with little emphasis on the role of right ventricular (RV) function. RV function has attracted increasingly more clinical attention in recent years. It has been shown that approximately 4–50% of HFpEF patients develop RV dysfunction (RVD), which is an independent risk factor for poor prognosis in this condition [95,96,97,98]. RVD is associated with clinical symptoms and echocardiographic findings in patients with HFpEF and provides a valuable contribution to the poor prognosis [97,99]. Therefore, the assessment of RV function in the HFpEF population is particularly important in clinical practice. HFpEF can result in RVD through a variety of mechanisms [100]. RV function is susceptible to afterload. In the setting of HFpEF, the reverse transmission of left heart pressure to the pulmonary veins caused by LV diastolic dysfunction increased LV filling pressure, reduced LA compliance, and elevated LA pressures, leading to the passive elevation of pulmonary pressure, elevation of RV afterload, and ultimately RVD [101]. It is well known that the systolic performance of left and right ventricles influences each other through the shared ventricular septum, and approximately 20% to 40% of RV contraction performance is derived from LV contraction, so systolic ventricular interdependence can directly or indirectly cause RVD [101,102]. In the setting of subtle LV systolic dysfunction in the HFpEF population, the direct contribution of left-to-right ventricular systolic function is reduced [100,102]. Several studies have found that atrial fibrillation (AF) is highly prevalent in the HFpEF population and is linked to RVD [99,100,103,104]. Both may aggravate HFpEF, in which elevated LV filling pressure leads to LA remodeling and dysfunction, and further to elevated pulmonary pressure and RV afterload—the process can be accelerated, especially with the development of AF [105]. Correspondingly, AF might directly damage RV function by reducing the RV longitudinal systolic performance and disrupting the rhythm of the myocardial fiber contraction [102]. In HFpEF patients with tricuspid regurgitation, RV volume overload leads to progressive RV remodeling, further contributing to RVD. Several HFpEF comorbidities including hypertension, diabetes mellitus, renal insufficiency, and obesity can lead to RV remodeling and dysfunction via metabolic dysfunction or systemic inflammation [102].

In summary, the accurate assessment of RV function is of great clinical significance. The localization of the RV posterior to the sternum and its complex morphology makes it challenging to accurately quantify RV function in routine clinical practice [97]. At present, CMR remains the gold standard for assessing RV function, but it is expensive; time-consuming to analyze; and cannot be used for patients with metal devices and claustrophobia, limiting its clinical application. Echocardiography is usually the preferred method for RV function evaluation because it is simple, inexpensive, used in real-time, non-invasive, and can be used at the bedside. A multiparametric approach is now recommended for RV function quantification by echocardiography, including tricuspid annular plane systolic excursion (TAPSE), tricuspid lateral annular systolic velocity wave (S’), and fractional area change (FAC) [106]. However, traditional RV function parameters are affected by load and angle dependence and do not allow for an accurate and comprehensive assessment of RV function. With the development and application of new echocardiographic techniques, the ability of echocardiography to assess RV function has been improved [107]. STE overcomes the limitations of traditional parameters and allows for the objective quantification of the RV function with high correlation between STE and CMR measurements [108]. The RV strain has been shown to be an accurate tool for the assessment of RV function and the identification of HFpEF patients with a high risk of adverse events [97].

In a prospective study with 201 HFpEF patients and 364 asymptomatic left ventricular diastolic dysfunction patients, RV function was obtained by 2D-STE; the researchers found the reductions in RV global longitudinal strain (RVGLS) and early diastolic longitudinal strain rate in patients with HFpEF. Moreover, the investigators revealed that RV systolic and diastolic dysfunction may contribute to the symptoms of HFpEF patients. Multiple regression analysis showed LVGLS to be an independent predictor of RV longitudinal systolic and diastolic function, emphasizing LV and RV diastolic and longitudinal systolic dysfunction as characteristics of HFpEF [109]. Subsequently, the team conducted a multicenter study, which confirmed that RVGLS and RV free wall longitudinal strain (RVFWLS) were significantly reduced in patients with HFpEF compared to healthy controls and patients with asymptomatic left ventricular diastolic dysfunction, but were less impaired than in patients with HFrEF. Simultaneously, RV strain parameters were reduced in HFpEF patients prior to changes in traditional RV functional parameters, suggesting that strain parameters are sufficiently sensitive to detect subtle RV dysfunction. In addition, RV strain parameters can predict clinical symptoms (dyspnea and NYHA ≥ III) in HFpEF patients, with better predictive ability than TAPSE, RVFAC, or S’ [110]. Lejeune et al., revealed that RVGLS had the highest correlation with CMR-RVEF (r = −0.617, *p* < 0.001) among all RV function parameters. Univariate and multivariate regression analyses showed that RVGLS was an independent predictor of HF hospitalization and all-cause mortality in HFpEF patients, providing incremental prognostic value over conventional RV function and clinical parameters. Consequently, researchers recommend RVGLS as part of the routine assessment of RV function for the identification of patients at high risk of adverse clinical events [97]. Due to the unique crescent shape of the right ventricle, 3D-STE is more suitable for RV function assessment than 2D-STE. Li et al., applied 3D-STE to assess RV functional changes in patients with HFpEF and to explore the prognostic value of 3D-STE parameters of the RV. The results indicated that 3D-RVFWLS was reduced in patients with HFpEF compared to controls, but there was no significant difference in RV volume and ejection fraction between the two groups. Survival analysis showed that 3D-RVFWLS was an independent predictor of poor prognosis in patients with HEpEF, and patients with 3D-RVFWLS < 22% had a significantly increased risk of adverse outcome [111]. Meng et al., analyzed RV systolic function in patients with HFpEF using both 2D-STE and 3D-STE and followed up the patients for a median time of 17 months. These researchers elucidated that 3D-RVFWLS, 3D-RVEF, and 2D-RVFWLS were all considered to be independent predictors of adverse outcomes in patients with HFpEF, and the predictive ability was comparable, supporting the ability of 3D-STE RV function parameters to identify patients with HFpEF with a high risk of adverse events [112].

### 3.4. Right Atrial Function

Right heart function is crucially important in HF patients, and RV function has been extensively studied, but little is known about the role of right atrium in HF. The right atrium plays an important role in RV filling and central venous pressure measurement. There are three components to right atrial (RA) function: reservoir function during atrial diastole; conduit function during passive emptying of right atrium; and booster function during active emptying of right atrium [113,114]. The causes of RA dysfunction are complex in patients with HFpEF. Pulmonary hypertension secondary to elevated LV filling pressures and LA dysfunction leads to RV remodeling and reduced diastolic compliance, further causing elevated RV filling pressures and thus elevated RA afterload [114,115]. In HFpEF patients with tricuspid insufficiency, regurgitation may cause RV dilation, RV dysfunction, and increased RA volume load [116]. AF is common in HFpEF patients, which has a direct impact on RA function through the induction of myocardial fibrosis and remodeling [104]. Previous studies have shown that RA size can characterize right heart function, and HFpEF patients with RA enlargement are at increased risk of adverse outcomes. Ikoma et al., showed a higher incidence of AF, more severe pulmonary hypertension, and worse RV systolic function in HFpEF patients with RA enlargement compared to patients with normal RA size. Additionally, patients with RA dilation had a significantly increased risk of adverse outcomes (HF hospitalization and all-cause death) [117]. Jain et al., comprehensively evaluated the RA function in HFpEF patients using RA strain and strain rate derived from CMR feature tracking technology. This prospective study suggested that RA reservoir function in HFpEF patients was impaired, while booster pump function was preserved. RA reservoir function and conduit function, rather than booster function, can independently predict all-cause mortality [113]. Conversely, Melenovsky et al., measured RA function with 2D-STE and found that RA reservoir and booster function were both impaired in HFpEF patients [100]. The different severity of HFpEF patients included in the two studies may explain the conflicting results. Recently, Kazuki et al., applied 2D-STE to a retrospective study of 108 control subjects and 89 HFpEF patients. The researchers found that compared with healthy controls, HFpEF patients have impaired RA reservoir and booster function during exercise, and these abnormal changes were related to decreased cardiac output, RVD, and poor aerobic capacity. Exercise-induced RA strain can provide incremental diagnostic value for HFpEF over LV and RV strain [118]. The above studies on CMR feature tracking technology, conventional echocardiography, and 2D-STE all indicated a certain degree of impaired RA function in patients with HFpEF, but there is still a lack of studies assessing the RA function of HFpEF patients by 3D-STE. In addition, the results of CMR and 2D-STE studies need to be verified by 3D-STE studies.

## 4. Summary and Prospect

At present, the assessment of cardiac function in patients with HFpEF by STE continues to attract increasing clinical attention. STE can detect subclinical cardiac dysfunction in patients with HFpEF early and sensitively, and the strain parameters obtained by STE are independent predictors of poor prognosis. Therefore, STE has important clinical value for risk stratification, disease severity assessment, treatment decision making, and prognosis estimation in patients with HFpEF. Most studies have recommended strain measurement as a part of routine echocardiography, However, some studies cited in this review are case–control in design. Consequently, the diagnostic parameters of a test derived from such a patient population may be less accurate. Considering that strain parameters are not routinely measured in clinical practice due to the lack of clear cut-off values and time-consuming measurements, our future studies will endeavor to how to make better use of STE and guide clinical practice.

## Figures and Tables

**Figure 1 diagnostics-13-02923-f001:**
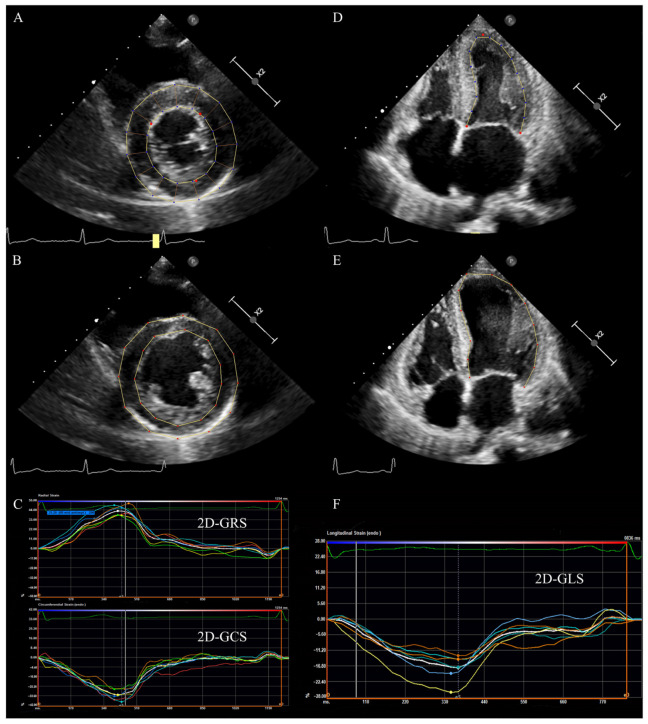
LV 2D strain in a patient with HFpEF. (**A**) LV short axis view at papillary muscle level: tracking LV endocardium and epicardium at end-systole. (**B**) LV short axis view at papillary muscle level: tracking LV endocardium and epicardium at end-diastole. (**C**) The radial and circumferential time–strain curves of LV short axis view. (**D**) LV apical four chamber view: tracking LV endocardium at end-systole. (**E**) LV apical four-chamber view: tracking LV endocardium at end-diastole. (**F**) The longitudinal time–strain curves of LV apical four-chamber view. LV: left ventricular; 2D: two-dimensional; GLS: global longitudinal strain; GRS: global radial strain, GCS: global circumferential strain, HFpEF: heart failure with preserved ejection fraction.

**Figure 2 diagnostics-13-02923-f002:**
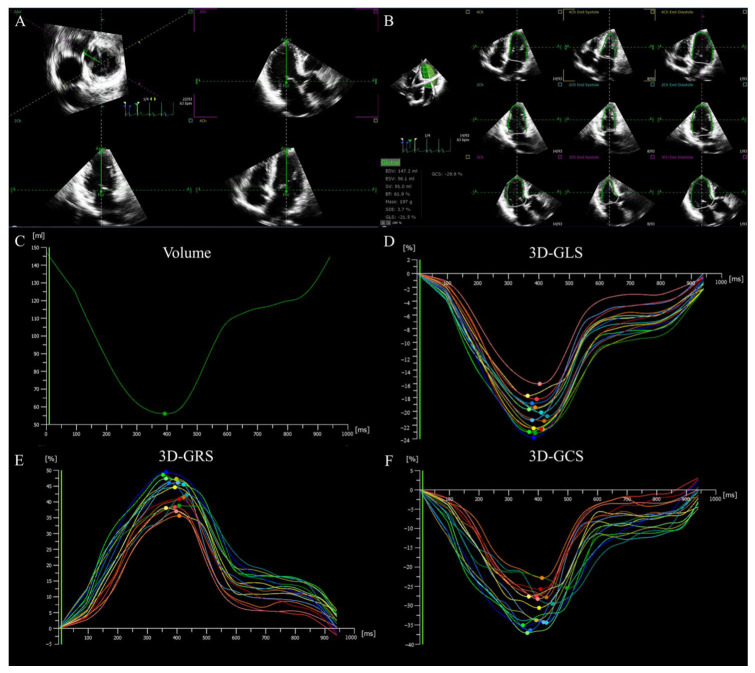
LV 3D strain in a patient with HFpEF. (**A**) Setting reference lines. (**B**) Identifying and tracking LV endocardium at end-diastole and end-systole. (**C**) LV time–volume curve. (**D**–**F**) LV longitudinal, radial and circumferential time–strain curves. LV: left ventricular; 3D: three-dimensional; GLS: global longitudinal strain; GRS: global radial strain, GCS: global circumferential strain, HFpEF: heart failure with preserved ejection fraction.

## Data Availability

Not applicable.

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
