# Peer review of "Clinical Usefulness of Speckle-Tracking Echocardiography in Patients with Heart Failure with Preserved Ejection Fraction"

_diagnostics, 2023, doi:10.3390/diagnostics13182923_

Round 1
Reviewer 1 Report
Lin et al. conducted a narrative review of speckle-tracking echocardiography in patients with heart failure with preserved ejection fraction (HFpEF). The authors provided a comprehensive overview of myocardial strain imaging (two-dimensional and three-dimensional speckle-tracking echocardiography), advantages, and limitations of strain imaging modalities. The manuscript explained the pathophysiology of HFpEF and how it affects left ventricle, left atria, right ventricle, and right atria. Furthermore, the authors described HFpEF changes in cardiac chambers which can be detected earlier by strain imaging, and their implications.
Concerns:
- The manuscript described the utility of strain imaging in garden variety HFpEF (secondary to hypertension, metabolic syndrome etc). The manuscript doesn’t mention other cardiac conditions that present as HFpEF but has distinct pathophysiology, for example, infiltrative, hypertrophic, and amyloid myocardial diseases. Authors should acknowledge other distinct pathophysiology of HFpEF, and if it is outside the scope of the current manuscript, it should be stated as such or address other pathophysiologies.
- Page 2, line 11- Can authors address how early and accurate identification of HFpEF would improve prognosis? Is there any study evaluating improved prognosis with early diagnosis? Is there any effective therapy improving prognosis in HFpEF patients?
- Similar to point number 2- page 5, line 20- how would it help in clinical management?
- Page 6, line 4- Kosmala et al.- Similar to Kosmala et al. (other examples Santos et al. (page 6 second line from bottom) and citation 83), most of the studies cited in the manuscript are case-control in design. Meaning, a study evaluating a diagnostic test has study population consist of cases and normal control. Diagnostic parameters of a test derived from such a patient population are erroneous. Such study design is one of the major limitations of a study evaluating diagnostic tests. This limitation needs to be stated in the limitation or conclusion section.
- Page 7, line 2, citation 81- In addition, they reported that patients with lower LA peak strain had higher rate of heart failure hospitalization and worse left ventricular systolic function, indicating that LA dysfunction is related to the severity and pathophysiology of HFpEF patients. – How systolic dysfunction findings were translated to HFpEF patients?
- Page 8, 10th line from bottom up- Consequently, researchers recommend RVGLS as part of the daily assessment of RV function for identification of patients at high risk for adverse clinical events. -Routine instead of daily.
- The summary and prospect section is appropriate.
Minor revision
Author Response
Point 1: The manuscript described the utility of strain imaging in garden variety HFpEF (secondary to hypertension, metabolic syndrome etc.). The manuscript doesn’t mention other cardiac conditions that present as HFpEF but has distinct pathophysiology, for example, infiltrative, hypertrophic, and amyloid myocardial diseases. Authors should acknowledge other distinct pathophysiology of HFpEF, and if it is outside the scope of the current manuscript, it should be stated as such or address other pathophysiologies.
Response 1: We appreciate the constructive suggestion. We agree with your viewpoints. By consulting the relevant literature, we have added subtypes of HFpEF caused by various causes to the manuscript.1 (Page 1)
Reference:
- Ge J. Coding proposal on phenotyping heart failure with preserved ejection fraction: A practical tool for facilitating etiology-oriented therapy. Cardiology journal. 2020; 27:97-98.
Point 2: Page 2, line 11- Can authors address how early and accurate identification of HFpEF would improve prognosis? Is there any study evaluating improved prognosis with early diagnosis? Is there any effective therapy improving prognosis in HFpEF patients?
Response 2: Thank you very much for your comments. HFpEF is a heterogeneous disease with complex etiology and pathophysiology. Early identification of HFpEF is helpful to find the risk factors, etiology and coexisting diseases of HFpEF and intervention for some specific causes of HFpEF can improve the prognosis. Previous study has shown that many HFpEF patients were only diagnosed at the first hospitalization for decompensated HF. Compared with outpatients with early diagnosis of HFpEF, HFpEF patients with a history of HF hospitalization showed worse renal function and higher events rates of COPD and anemia, which may be due to delayed diagnosis leading to delayed treatment intervention and worse clinical outcomes.1 In addition, Saito et al. found that early diagnosis of HFpEF patients can be made by exercise stress testing. Then early treatment of these patients through guideline-recommended therapies may reduce the incidence of adverse outcomes in patients.2 Therefore, HFpEF patients should be identified and evaluated as early as possible, and follow the relevant guidelines for treatment and management.3 A large number of previous studies have shown that Sodium–glucose cotransporter 2 inhibitors can reduce the risk of hospitalization for heart failure in HFpEF patients.4-7 In addition, a variety of drugs such as Sacubitril/Valsartan andmineralocorticoid receptor antagonist can reduce the risk of heart failure hospitalization in HFpEF patients.8,9
- Reddy YNV, Obokata M, Jones AD, et al. Characterization of the Progression From Ambulatory to Hospitalized Heart Failure With Preserved Ejection Fraction. J Card Fail. 2020; 26: 919–928.
- Saito Y, Obokata M, Harada T, et al. Prognostic benefit of early diagnosis with exercise stress testing in heart failure with preserved ejection fraction. Eur J Prev Cardiol. 2023; 30: 902-911.
- Ge J. Coding proposal on phenotyping heart failure with preserved ejection fraction: A practical tool for facilitating etiology-oriented therapy. Cardiology journal. 2020; 27:97-98.
- Anker SD, Butler JPacker M. Empagliflozin in heart failure with a preserved ejection fraction. reply. N Engl J Med. 2022; 386(21): e57.
- Bayes-Genis A, Aimo ALupón J. Empagliflozin in heart failure with preserved and mildly reduced ejection fraction: prognostic benefit confirmed with different endpoint definitions. Eur J Heart Fail. 2022; 24(8): 1406-1409.
- Solomon SD, Vaduganathan M, Claggett BL, et al. Baseline characteristics of patients with HF with mildly reduced and preserved ejection fraction: DELIVER trial. JACC Heart Fail. 2022; 10(3): 184-197.
- Anker SD, Siddiqi TJ, Filippatos G, et al. Outcomes with empagliflozin in heart failure with preserved ejection fraction using DELIVER-like endpoint definitions. Eur J Heart Fail. 2022; 24(8): 1400-1405.
- Solomon SD, Vaduganathan M, B LC, et al. Sacubitril/valsartan across the spectrum of ejection fraction in heart failure. Circulation. 2020; 141(5): 352-361.
- Pfeffer MA, Claggett B, Assmann SF, et al. Regional variation in patients and outcomes in the treatment of preserved cardiac function heart failure with an aldosterone antagonist (TOPCAT) trial. Circulation. 2015; 131(1): 34-42.
Point 3: Similar to point number 2- page 5, line 20- how would it help in clinical management?
Response 3: Thank you very much for your comments. After early identification of HFpEF, intervention should be first carried out according to different etiologies. On this basis, then early combined drug therapy and regular follow-up assessment. Long-term management of HFpEF patients is emphasized, including the management of risk factors and comorbidities, exercise rehabilitation, enhancement of patient education, and improvement of adherence to treatment.1 Based on the etiological classification of HFpEF, different HFpEF subtypes have different management recommendations.1
- Chinese Expert Consensus Working Group on Diagnosis and Treatment of Heart Failure With Preserved Ejection Fraction. Diagnosis and Treatment of Heart Failure With Preserved Ejection Fraction: Chinese Expert Consensus 2023. Chinese Circulation Journal.2023; 8(4): 375-393.
Point 4: Page 6, line 4- Kosmala et al.- Similar to Kosmala et al. (other examples Santos et al. (page 6 second line from bottom) and citation 83), most of the studies cited in the manuscript are case-control in design. Meaning, a study evaluating a diagnostic test has study population consist of cases and normal control. Diagnostic parameters of a test derived from such a patient population are erroneous. Such study design is one of the major limitations of a study evaluating diagnostic tests. This limitation needs to be stated in the limitation or conclusion section.
Response 4: We thank the reviewer for this valuable comment. We agree with your viewpoints. We have added the limitations of case-control studies to the summary section.
Point 5: Page 7, line 2, citation 81 (90)- In addition, they reported that patients with lower LA peak strain had higher rate of heart failure hospitalization and worse left ventricular systolic function, indicating that LA dysfunction is related to the severity and pathophysiology of HFpEF patients. – How systolic dysfunction findings were translated to HFpEF patients?
Response 5: Thank you very much for your comments. We read the reference cited in this sentence carefully again. the study by Santos et al. was conducted on HFpEF patients to investigate changes in LA function and to sought to determine the clinical and echocardiographic correlates of reduced LA strain in HFpEF patients. The results of this study showed that among HFpEF patients grouped by the quartiles of systolic LA strain, those with lower LA strain had a higher prevalence of prior heart failure hospitalization as well as worse LV systolic function (measured by LVEF and LV global longitudinal strain). The lower LA strain, the smaller LVEF and the absolute value of GLS, but the LVEF was within the normal range. Therefore, LA strain reduction is associated with relatively worse left ventricular systolic function rather than systolic dysfunction.
Point 6: Page 8, 10th line from bottom up- Consequently, researchers recommend RVGLS as part of the daily assessment of RV function for identification of patients at high risk for adverse clinical events. -Routine instead of daily.
Response 6: Thanks for your careful review. We also agree with your comment and have replaced daily with routine in the manuscript.
Reviewer 2 Report
The authors have conducted a comprehensive review of the most relevant articles regarding the use of speckle-tracking in HFpEF, effectively analyzing the most important aspects of the function of both atria and ventricles. The whole paper is well organized and the references have been carefully chosen.
As for the classification of heart failure based on the ejection fraction, I would recommend delving into the difficulty of using this parameter to define the complexity of this clinical syndrome:
- “Heart Failure Pharmacological Management: Gaps and Current Perspectives.” Journal of clinical medicine vol. 12,3 1020. 28 Jan. 2023, doi:10.3390/jcm12031020
- “Do the Current Guidelines for Heart Failure Diagnosis and Treatment Fit with Clinical Complexity?.” Journal of clinical medicine vol. 11,3 857. 6 Feb. 2022, doi:10.3390/jcm11030857
The review is primarily focused on the role of echocardiography. However, in the introduction, I would further emphasize the role of cardiac magnetic resonance imaging in characterizing HFpEF:
- “Myocardial deformation assessed among heart failure entities by cardiovascular magnetic resonance imaging.” ESC heart failure vol. 8,2 (2021): 890-897. doi:10.1002/ehf2.13193.
- “Myocardial Tissue Characterization in Heart Failure with Preserved Ejection Fraction: From Histopathology and Cardiac Magnetic Resonance Findings to Therapeutic Targets.” International journal of molecular sciences vol. 22,14 7650. 17 Jul. 2021, doi:10.3390/ijms22147650.
- “CMR in the Evaluation of Diastolic Dysfunction and Phenotyping of HFpEF: Current Role and Future Perspectives.” JACC. Cardiovascular imaging vol. 13,1 Pt 2 (2020): 283-296. doi:10.1016/j.jcmg.2019.02.031
Finally, regarding the role of strain in HFpEF, I would deepen the prognostic value of left atrial strain:
- “Prognostic Utility and Clinical Significance of Cardiac Mechanics in Heart Failure With Preserved Ejection Fraction: Importance of Left Atrial Strain.” Circulation. Cardiovascular imaging vol. 9,3 (2016): 10.1161/CIRCIMAGING.115.003754 e003754. doi:10.1161/CIRCIMAGING.115.003754
the proposed references cover some important aspects which are not mentioned in their review, particularly on limits of LVEF, and the importance of CMR and left atrial strain, showing papers where they can find those suggested aspects and, then, enrich the review.
If Authors will add these (or other) references, it would be at their own discretion.The use of the English language is good both in terms of form and syntax. I would recommend proofreading the text for the orthographic correction of some words and abbreviations
Author Response
As for the classification of heart failure based on the ejection fraction, I would recommend delving into the difficulty of using this parameter to define the complexity of this clinical syndrome:
- “Heart Failure Pharmacological Management: Gaps and Current Perspectives.” Journal of clinical medicine vol. 12,3 1020. 28 Jan. 2023, doi:10.3390/jcm12031020
- “Do the Current Guidelines for Heart Failure Diagnosis and Treatment Fit with Clinical Complexity?.” Journal of clinical medicine vol. 11,3 857. 6 Feb. 2022, doi:10.3390/jcm11030857
The review is primarily focused on the role of echocardiography. However, in the introduction, I would further emphasize the role of cardiac magnetic resonance imaging in characterizing HFpEF:
- “Myocardial deformation assessed among heart failure entities by cardiovascular magnetic resonance imaging.” ESC heart failure vol. 8,2 (2021): 890-897. doi:10.1002/ehf2.13193.
- “Myocardial Tissue Characterization in Heart Failure with Preserved Ejection Fraction: From Histopathology and Cardiac Magnetic Resonance Findings to Therapeutic Targets.” International journal of molecular sciences vol. 22,14 7650. 17 Jul. 2021, doi:10.3390/ijms22147650.
- “CMR in the Evaluation of Diastolic Dysfunction and Phenotyping of HFpEF: Current Role and Future Perspectives.” JACC. Cardiovascular imaging vol. 13,1 Pt 2 (2020): 283-296. doi:10.1016/j.jcmg.2019.02.031
Finally, regarding the role of strain in HFpEF, I would deepen the prognostic value of left atrial strain:
- “Prognostic Utility and Clinical Significance of Cardiac Mechanics in Heart Failure With Preserved Ejection Fraction: Importance of Left Atrial Strain.” Circulation. Cardiovascular imaging vol. 9,3 (2016): 10.1161/CIRCIMAGING.115.003754 e003754. doi:10.1161/CIRCIMAGING.115.003754
the proposed references cover some important aspects which are not mentioned in their review, particularly on limits of LVEF, and the importance of CMR and left atrial strain, showing papers where they can find those suggested aspects and, then, enrich the review.
Response: We appreciate the constructive suggestion. We have read your recommended literatures and enriched our review in the appropriate places in the manuscript. The new literatures cited in the review have been marked purple in the references part.